# A Mutation in DNA Polymerase α Rescues *WEE1*^KO^ Sensitivity to HU

**DOI:** 10.3390/ijms22179409

**Published:** 2021-08-30

**Authors:** Thomas Eekhout, José Antonio Pedroza-Garcia, Pooneh Kalhorzadeh, Geert De Jaeger, Lieven De Veylder

**Affiliations:** 1Department of Plant Biotechnology and Bioinformatics, Ghent University, 9052 Gent, Belgium; thomas.eekhout@psb.vib-ugent.be (T.E.); joseantoniopedrozag@gmail.com (J.A.P.-G.); poonehkalhor@yahoo.com (P.K.); gejae@psb.vib-ugent.be (G.D.J.); 2Center for Plant Systems Biology, VIB, 9052 Gent, Belgium

**Keywords:** replication stress, DNA damage, cell cycle checkpoint

## Abstract

During DNA replication, the WEE1 kinase is responsible for safeguarding genomic integrity by phosphorylating and thus inhibiting cyclin-dependent kinases (CDKs), which are the driving force of the cell cycle. Consequentially, *wee1* mutant plants fail to respond properly to problems arising during DNA replication and are hypersensitive to replication stress. Here, we report the identification of the *polα-2* mutant, mutated in the catalytic subunit of DNA polymerase α, as a suppressor mutant of *wee1*. The mutated protein appears to be less stable, causing a loss of interaction with its subunits and resulting in a prolonged S-phase.

## 1. Introduction

DNA replication is a highly complex process that ensures the chromosomes are correctly replicated to be passed onto the daughter cells during mitosis. Replication starts at multiple different origins throughout the genome, but in contrast to *S. cerevisiae* where origins are characterized by a specific sequence, origins in plants appear to be rather specified by a certain epigenetic state [1]. Origin firing, the process of activating an origin, is strictly controlled by cyclin-dependent kinases (CDKs) in complex with their regulatory cyclin subunits. A plethora of proteins are involved in the process of DNA replication, including PCNA, MCMs, DNA polymerase α, δ, and ε, chromatin assembly factors, and others (for a recent review, see [2]). Among the polymerases, DNA polymerase α is a protein complex consisting of a catalytic subunit, an accessory subunit, and two RNA primases. The primase subunits are responsible for initiating replication at both leading and lagging strands by synthesizing around 10 nucleotides of RNA primer. After this, the primase subunit is displaced by the catalytic polymerase subunit and another 20–30 deoxyribonucleotides are synthesized [3,4]. However, the DNA polymerase α processivity is very low and its exonuclease domain is inactive, causing it to have no proofreading capability. Until recently, it was thought that DNA polymerase α is replaced rather quickly by DNA polymerase ε on the leading strand and DNA polymerase δ on the lagging strand [5]. Recent work, however, suggests that DNA polymerase δ synthesizes both strands, whereas DNA polymerase ε is rather involved in removing replication errors of polymerase δ and playing a scaffolding role at the replication fork [6].

Next to their role in DNA synthesis, over the years it has become increasingly clear that polymerases play a role in epigenetic inheritance in plants. In genetic screens looking for repressors of DNA methylation-dependent gene silencing, different mutant alleles of all three polymerases have been identified [7,8,9,10]. Two reported mutants in polymerase α, called *incurvata2-1* and *polα-1*, display similar phenotypes, being smaller leaves and early flowering [8,11], which were shown to originate from loss of epigenetic repression of flowering-specific genes. Polymerase α interacts with TFL2/LHP1, a protein responsible for depositing heterochromatin marks on histones, and in this way, it could help to maintain repressive histone marks at the replication fork. Similarly, missense mutants of polymerases δ and ε display, next to increased sensitivity to certain kinds of DNA damage, accelerated flowering that is correlated with changes in heterochromatin marks such as H3K27me3 and H3K4me3 [7,9,12,13]. This is similar to the situation in fission yeast, where the catalytic subunit of polymerase α binds to the LHP1 homolog in order to maintain heterochromatin structure by recruiting histone methyltransferases [14,15].

To safeguard genomic integrity and allow faithful transmission of the DNA to their progeny, eukaryotes have developed extensive mechanisms and signaling pathways that control the fidelity of the DNA replication process. In yeast and mammals, where this process has been widely studied due to its link with oncogenesis, there are two serine/threonine protein kinases called ATAXIA TELANGIECTASIA MUTATED (ATM) and ATAXIA TELANGIECTASIA MUTATED AND RAD3-RELATED (ATR) that are involved in the sensing and responding to double-stranded breaks (DSBs) and single-stranded DNA (ssDNA), respectively. Downstream of these two sensor kinases two checkpoint kinases, Chk1 and Chk2, act upon a plethora of downstream targets, including the p53 transcription factor that induces expression of DNA repair genes, cell cycle regulators, and possible cell death-inducing genes [16]. In plants, the upstream ATM and ATR kinases are conserved, but the checkpoint kinases Chk1 and Chk2 and the transcription factor p53 are missing, although plants have a functional homolog called SUPPRESSOR OF GAMMA RESPONSE 1 (SOG1) [17].

During replication, the most common type of damage that occurs is a stalled replication fork, which can arise at any point where the progress of the DNA processing enzymes is blocked. Endogenous causes include bulky nucleotides (e.g., through alkylation), shortage of dNTPs, crosslinked DNA, protein–DNA adducts, etc. Exogenous causes of stalled replication forks are mostly pyrimidine dimers caused by UV radiation. A stalled replication fork leads to a stretch of single-stranded DNA (ssDNA) being unwound by the uncoupling of the helicase from the replisome, resulting in activation of ATR [18]. This activates a signaling cascade called the intra-S replication checkpoint that, amongst others, causes stabilization of the stalled replication forks [19]. Next to stabilizing existing replication forks, the replication checkpoint also slows down the firing of new origins, a process in humans that is dependent on WEE1-mediated CDK inhibition [20]. WEE1 is a tyrosine kinase, which phosphorylates CDKs to inhibit their activity. In the absence of *WEE1*, new replication forks continue to be initiated and will then stall and collapse, leading to double-strand breaks (DSBs) [20]. In *Arabidopsis*, both *ATR* and *WEE1* are necessary to survive replication stress [21,22], with mutants showing drastically shortened roots when grown on medium containing chemicals that induce replication stress, such as hydroxyurea (HU). Here, we report the identification of a suppressor mutation in the gene encoding the catalytic subunit of *Arabidopsis* DNA polymerase α able to partially rescue root growth of *wee1* plants on HU-containing medium. We furthermore show that this mutation has an effect on the duration of the DNA replication phase, likely by limiting the availability of functional DNA polymerase α complexes.

## 2. Results

### 2.1. A Mutation in the Catalytic Subunit of Polymerase α Rescues HU Sensitivity of WEE1^KO^ Plants

When *WEE1*^KO^ plants are grown on a medium containing HU, they show strong inhibition of root growth [22]. An ethyl methanesulfonate (EMS) mutagenesis screen was performed to identify mutations that can rescue this root growth inhibition. One of the revertants found was line 113–2, which showed improved root growth on HU compared to wee1-1 plants. By outcrossing this line to a wee1-1 mutant line in Landsberg erecta background, a mapping population was obtained that could be used for next-generation sequencing-based mutation mapping using the SHORE pipeline coupled to the SHOREmap algorithm [23]. The causal mutation of line 113–2 was mapped to a genomic region on the lower arm of chromosome 5, more exactly to the *At5G67100* gene, where it leads to the nonsynonymous P846S amino acid change (Figure 1A). The mutation was confirmed in the original 113–2 line by direct Sanger sequencing (Appendix A). The *At5G67100* gene, also known as *INCURVATA2* (*ICU2*), encodes the catalytic subunit of the DNA polymerase α, therefore we named this new mutant allele *polα-2*. When modeled to the human polymerase α structure [24], the *polα-2* mutation appears to be located on the border of the hinge between the N-terminal domain (yellow in Figure 1A,B) and the palm domain (green in Figure 1A,B), which is an unstructured and thus uncharacterized region of the protein (grey in Figure 1A), but the protein sequence of the region directly surrounding the mutation is conserved in all eukaryotes (Figure 1C).

Because T-DNA insertion lines of the DNA polymerase α gene are lethal, we opted for a complementation strategy to prove that the identified allele indeed corresponds to the causative mutation rescuing *wee1-1* hypersensitivity to HU. To this end, we generated a construct containing the complete coding sequence of wild-type *ICU2*, driven by the 2 kb endogenous promoter, and transformed this into the original 113–2 line. After selecting a single-insert line, we grew this complementation line on HU and indeed noticed that this line regained sensitivity to HU (Appendix A). It also indicates that the P846S mutation is not a dominant gain-of-function mutation, since the presence of both WT and mutated polymerase α results in wee1 sensitivity to HU.

After crossing out the *wee1-1* mutation, root growth of both single and double mutant was compared on control medium and medium supplemented with HU. Under control conditions, both *polα-2* and *polα-2 wee1-1* have shorter roots compared to Col-0 (Figure 1D,F). When germinated on an HU-containing medium, the *polα-2 wee1-1* mutant partially rescues root growth of *wee1-1* but has shorter roots than the *polα-2* single mutant (Figure 1E,F). This indicates that the mutation in *polα-2* already leads to some kind of growth penalty under control conditions that is not WEE1-dependent since the double mutant shows the same reduction in root length.

### 2.2. The Polα-2 Mutant Is Distinct from Earlier Described Polymerase Mutants

A previously reported mutation in the *INCURVATA2* gene, *icu2-1*, has been described as having curled leaves (hence the name), which is caused by the epigenetic deregulation and thus ectopic expression of flowering genes such as *APETALA3* (*AP3*), *SEPALATA3* (*SEP3*), and *FLOWERING LOCUS T* (*FT*) in the leaves [11]. A very similar phenotype is attributed to another allele of polymerase α, *polα-1* [8]. When these mutations are modeled to the human polymerase α structure, they are both located in the thumb domain (Figure 1C). We compared the *icu2-1* mutant (Landsberg erecta background) leaf phenotype with that of the *polα-2* mutant under control conditions. In contrast to *icu2-1*, a curled leaf phenotype is absent in the *polα-2* mutant (Figure 2A). Moreover, as reported before [11], flowering genes such as *AP3*, *SEP3*, and *FT* are significantly upregulated in the leaves of the *icu2-1* mutant, but not in the *polα-2* mutant (Figure 2B).

### 2.3. The Polα-2 Mutation Causes a Prolongation of the S-Phase and Activation of the DNA Damage Response

To investigate if the mutation in *polα-2* has an influence on cell cycle progression, we estimated the duration of the S-phase and cell cycle using 5-ethynyl-2′-deoxyuridine (EdU) labeling [25]. Remarkably, the S-phase duration of *polα-2* lasted more than twice as long as in Col-0. A similar increase was observed in the *polα-2 wee1-1* double mutant, indicating that this prolongation of the S-phase is WEE1 independent (Figure 3A). The extension of the S-phase by approximately 4 h alone cannot explain the 10 h increase in total cell cycle duration, indicating the activation of a second cell cycle arrest, possibly in the G2/M-phase (Figure 3B) as previously reported for *polα-1* [8].

When plants were treated with 1 mM HU, Col-0 plants showed an extended S-phase duration that is not observed in the *wee1-1* plants, consistent with previous reports of the role of WEE1 in arresting the S-phase during the replicative stress response [26,27]. In the *polα-2* plants, the HU treatment resulted only in an extended cell cycle duration, whereas in the *polα-2 wee1-1* plants both S phase and total cell cycle length increased (Figure 3). Overall, these data indicate that impairment of Pol α in the *polα-2* mutant leads to mainly a WEE1-independent S-phase arrest, which allows coping with HU-induced replication defects. Nevertheless, under replication stress conditions, the presence of WEE1 contributes that *pol**α-2* plants proceed through S-phase.

Additionally, in Col-0 and *polα-2* untreated root tips, we evaluated the expression of DDR-related genes through qRT-PCR. All genes associated with DNA repair (*BRCA1*, *RAD51*, *PARP2*, *RAD17* and *TSO2*) were upregulated in *polα-2* compared to Col-0, indicating the presence of endogenous DNA damage in *polα-2* (Figure 4). Among the tested DNA-damage responsive cell cycle inhibitors *SMR4*, *SMR5*, and *SMR7* [28], only *SMR7* was upregulated (Figure 4).

### 2.4. Polα-2 Cannot Rescues HU Sensitivity of ATR^KO^ and SOG1^KO^ Plants

To know whether other main DDR regulators as ATM, ATR, and SOG1 contribute to the pre-activation of the DDR in the *polα-2* mutant, we generated the corresponding double mutants to test their response to replication stress. Sensitivity to replication stress has been previously described for both *atr-2* and *sog1-1* mutants [21,29], as observed in this work (Figure 5A,C). Strikingly, while the *polα-2* mutation confers tolerance to replication stress in *wee1-1* mutant plants, no rescue of root growth under replication stress can be seen in *polα-2 atr-2* or *polα-2 sog1-1* double mutant lines (Figure 5A,C). Indeed, the *polα-2 sog1-1* double mutant displayed higher sensitivity to HU than *sog1-1* plants (Figure 5C). Furthermore, no differences in tolerance to replication stress could be observed in the *polα-2 atm-2* double mutant (Figure 5B), although statistical analysis indicates a partial rescue of root growth under control conditions back to WT length, suggesting that an ATM-induced G2/M-checkpoint might partially account for the shorter root of the *polα-2* mutant.

### 2.5. The Mutated Polymerase α Protein Is Unstable and Loses Interaction with the Other Subunits

The polymerase alpha is part of a multisubunit complex [4]. Because the mutation in *polα-2* resides in a loop structure (Figure 1), we were interested to see if this results in a potential loss of interaction partners. Therefore, both wild-type (POLA-WT) and mutant (POLA-mut) coding sequences were cloned and tagged N-terminally with the GSRhino-tag, a tandem affinity purification (TAP) tag. These constructs were then expressed under the control of the CaMV 35S promoter in *Arabidopsis* PSB-D cultured cells [30]. When the accumulation of the tagged proteins was tested, the wild-type protein was found to run on a protein gel at the expected height of 191 kDa. Differently, the mutant protein appeared in two distinct bands, one at the expected height and one around 60 kDa (Figure 6A). Since the TAP-tag is located at the N-terminus, this means that the mutated protein is partly truncated, possibly indicating reduced stability of the mutant form. The wild-type protein copurifies with the other subunits of the DNA polymerase α, being the noncatalytic subunit POLA2 and the two DNA primases POLA3 and POLA4. The mutant protein also copurifies with the other subunits, but these are found at a much lower frequency, likely due to the instability of the mutant polymerase protein (Figure 6B). While no other interactors are found with the mutant protein, the wild-type protein also interacts at a lower frequency with STN1 and CTC1, which are two telomere binding proteins that are known to interact with polymerase α [31], and with EOL1, a WD40-repeat-containing protein that interacts with LHP1-PCR2 to maintain epigenetic marks on histones [32] (Figure 6B).

## 3. Discussion

DNA polymerase α is not only essential to initiate DNA replication at both leading and lagging strands but also directly or indirectly interacts with chromatin remodeling complexes to re-establish the chromatin state after DNA replication [8,11]. Here, we report on the identification of a missense mutation in DNA polymerase α that is able to rescue the hypersensitive response of *WEE1*^KO^ plants to HU-induced replication stress. Since knockout mutations in polymerase α are lethal [11], the mutant allele isolated here might be either a partial loss-of-function or a gain-of-function, although the recessive nature of the mutation argues for the former. HU causes replication stress by inhibiting the RIBONUCLEOTIDE REDUCTASE (RNR) complex, in this way lowering the dNTP pool and stalling the polymerases at the replication fork [33]. Although replication is arrested, helicases likely continue to unwind DNA and, in this way, create long stretches of ssDNA which are, together with the nearby stalled replication fork, a substrate for the activation of ATR [18]. Strikingly, while a mutated polymerase α can rescue the strong phenotype of *wee1* plants on HU, DNA polymerase α appears to be necessary for the first steps of replication stress sensing, and more specifically its coupled primase activity is needed to activate the replication checkpoint mediated by ATR [34].

We showed that the proline-to-serine mutation occurs in a distinct domain from previously described mutations, being in the N-terminal domain instead of the thumb domain (Figure 1C). This might explain why, in contrast to the *icu2-1* mutant that loses chromatin-mediated repression of flowering genes in the leaves [11], the *polα-2* mutation does not show these changes in expression (Figure 2B). Furthermore, in contrast to *icu2-1* and *polα-1*, the *pol**α-2* plants do not show the curled leaf phenotype (Figure 2A) [8,11]. It is however difficult to estimate the consequence of the *polα-2* P846S mutation on the protein structure because it is located close to an unstructured loop region that is missing in the crystal structure of the human ortholog [24]. When the mutated protein, fused to an N-terminal tag, was expressed in cell cultures, a truncated form of around 60 kDa was visible next to the expected full-length protein (Figure 6A). Prolines are known to increase the rigidity of the protein backbone, which is why they are usually found at the beginning of loop structures, as is the case here. The mutation to a serine, which is one of the most flexible amino acids [35], would thus increase flexibility at a location where rigidity is needed, making the protein potentially unstable. The single truncated band instead of a protein smear hints at the specific breaking of the protein at a certain location instead of non-specific protein degradation. However, at least a part of the DNA polymerase α pool should still be functional, given that the plants are viable. The presence of a lower abundant full-length protein band supports this hypothesis (Figure 6A).

In yeast, it was shown that the N-terminal region of polymerase α is important to bind through Ctf4 to the CMG helicase at the replication fork [36]. In plants, ENHANCER OF LHP1 (EOL1) is a Ctf4-related protein that interacts with LHP1 [32] and is found in our TAP experiment, but only bound to the WT protein. We hypothesize that the truncated form cannot interact efficiently anymore with the replication fork, resulting in less available functional DNA polymerase α. Since DNA polymerase α is necessary to initiate replication at both leading and lagging strands, less availability of the protein means that replication will slow down, resulting in an impaired initiation of the replication.

Molecular analysis of root tips of the *polα-2* mutant indicated the presence of endogenous DNA damage that leads to a pre-activation of the DNA damage response (DDR) in these plants, characterized by the upregulated expression of genes involved in HR-dependent DNA repair such as *BRCA1* and *RAD51* [37,38], and by induction of the *SMR7* cell cycle inhibitor (Figure 4) [28]. An increase in the frequency of intrachromosomal homologous recombination has been described for multiple DNA polymerase mutants, often linked with an increase in HR-related gene expression [8,9,12,13]. Studies in other eukaryotes have shown that homologous recombination, and in particular the RAD51 protein, is necessary to stabilize stalled replication forks and, in a later stage, restart them [39]. SMR7 is known to respond to several types of DNA damage, both endogenous [27] and exogenous [28], and is induced to a similar extent in the polymerase ε mutant *abo4-1* [40].

Moreover, the impairment of Polα in the *polα-*2 mutant led to an increase in the length of the S-phase and total cell cycle duration in the *polα-2* mutant, which was independent of WEE1 (Figure 3) and likely causes the shorter root phenotype under control conditions (Figure 1D). Likely the slower progressivity of the polymerase and the accumulation of endogenous replication stress accounts for the longer S phase. The induction of the CDK inhibitor *SMR7* could be responsible for the possible G2/M-phase checkpoint that causes the longer total cell cycle duration (Figure 3). A very similar lengthening of both S-phase and total cell cycle duration has been described for mutants of the *Arabidopsis* DNA polymerase ε [40]. These *abo4* mutants were more tolerant to HU, suggesting a similar effect as the *polα-2* mutation. However, combination with *WEE1*^KO^ plants was embryo-lethal, showing that either the replication stress response is much stronger activated in these *abo4* mutants, or that another mechanism is involved.

We hypothesize that the mechanistic slowing down of S phase progression of the *polα-2* plants likely allows *wee1-1* mutants to cope with the reduced availability of dNTPs induced by HU application. The observation that the *polα-2* mutation can complement *wee1-1* but not *atr-2* and *sog1-1* mutants indicates that WEE1 predominantly coordinates the speed of DNA replication with the dNTP availability, a notion supported by the previous observations that *wee1-1* HU-hypersensitivity can be overcome as well by mutations in subunits of the RNase H2 complex likely by allowing substitution of rNTPs for dNTPs into the replicating DNA [41,42]. Differently, ATR and SOG1 likely also contribute to the stabilization of the stalled replication forks, explaining why *atr-2* and *sog1-1* mutants cannot be rescued by the *polα-2* mutation when grown under replication stress.

In conclusion, we report on the identification of a mutation in DNA polymerase α that is able to rescue *wee1-1* hypersensitivity to replication stress. The mutated polymerase causes a lengthening of S-phase duration that is independent of WEE1, likely by limiting the pool of available functional DNA polymerase alpha complexes.

## 4. Materials and Methods

### 4.1. Plant Materials and Growth Conditions

*Arabidopsis thaliana* plants (all Col-0 accession except for *icu2-1*, which is in Landsberg *erecta* background) were grown under long-day conditions (16 h/8 h) at 22 °C on ½ Murashige and Skoog medium containing 10 g/L sucrose, 0.5 g/L MES, pH 5.7 and 10 g/L agar for vertical growth or 8 g/L agar for horizontal growth. The *wee1-1, atr-2*, *atm-2* and *sog1-1* alleles were described previously [21,22,43,44]. For treatment with HU, seeds were plated directly on a control ½ MS medium or medium containing 0.75 mM HU. The root length of 7 DAS seedlings was measured. Primers used for genotyping are listed in Appendix A.

### 4.2. EMS Mutagenesis and Mapping

EMS mutagenesis has been described before [42]. Line 113–2 increased the resistance of *wee1* to HU and was crossed to the Landsberg erecta accession containing the *wee1-1* mutation. F1 plants were self-fertilized and the F2 population was screened for increased HU resistance. Leaf samples of around 200 plants were pooled and DNA was extracted using the DNeasy Plant Mini Kit (QIAGEN, Hilden, Germany) according to the manufacturer’s protocol. Illumina TruSeq libraries were generated and sequenced on an Illumina NextSeq 500 150bp paired-end run. We used the SHORE pipeline [45] to map the obtained reads to the Col-0 reference genome (TAIR10). SNPs were determined based on the alignment and the relative allele frequencies were compared between the two parental genomes (Col-0 and Ler) using SHOREmap [23].

### 4.3. RNA Extraction and qRT-PCR

Root tips (around 2 mm) of 7-day-old seedlings were collected in liquid nitrogen. RNA from samples was extracted using the RNeasy Plant Mini kit (QIAGEN) and cDNA was prepared from 1 µg of RNA using the iScript cDNA synthesis kit (Bio-Rad, Hercules, CA, USA), both according to the manufacturer’s protocols. Quantitative RT-PCR was performed in a final volume of 5 µL with SYBR Green I Master (Roche, Basel, Swiss) and analyzed with a Lightcycler 480 (Roche, Basel, Swiss). For each reaction, three biological and three technical repeats were done. Expression levels were normalized by the three reference genes EMB2386, RPS26C, and PAC1. Primers used for qRT-PCR are listed in Appendix A.

### 4.4. Tandem Affinity Purification and Western Blot

TAP-MS analysis was performed as described previously [30]. Briefly, the plasmids expressing Pola-WT and Pola-MUT fused to the double affinity GSrhino tag [30] were transformed into *Arabidopsis* (Ler) cell-suspension cultures. TAP purifications were performed with 200 mg of total protein extract as input and interacting proteins were identified by mass spectrometry using a Q Exactive (Thermo Scientific, Waltham, MA, USA) orbitrap. Proteins with at least two high-confidence peptides were retained only if reproducible in two experiments. Non-specific proteins were filtered out based on their frequency of occurrence in a large dataset of TAP experiments with many different and unrelated baits as described [30]. For Western blot analysis, 50 µg of total protein extract was separated in 4–15% Mini-PROTEAN TGX Precast Protein Gel (Bio-Rad, Hercules, CA, USA), transferred to a PVDF membrane using Trans-blot turbo transfer packs (Bio-Rad, Hercules, CA, USA), and membranes were incubated overnight at 4 °C with 3% skimmed milk 1x TBST solution. GSrhino-tagged proteins were detected with peroxidase anti-peroxidase soluble complex antibody (1/2500).

### 4.5. EdU Labelling

For cell cycle length analysis, we used a method adapted from [25]. Plants were grown on supplemented MS medium (10 g L^−1^ sucrose, 0.1 g L^−1^ myo-inositol, 0.5 g L^−1^ MES, 100 μL thiamine hydrochloride (10 mg mL^−1^), 100 μL pyridoxine (5 mg mL^−1^), 100 μL nicotinic acid (5 mg mL^−1^), pH 5.7, adjusted with 1 m KOH, and 10 g L^−1^ agar) for 5 days, and transferred to the same medium supplemented with EdU (10 µM). Samples were collected after 3, 6, 9, and 12 h, fixed in paraformaldehyde (4% in PME buffer: 50 mm piperazine-N,N′-bis (2-ethanesulphonic acid) (PIPES), pH 6.9; 5 mM MgSO4; 1 mM EGTA) for 45 min and washed with PME 1X buffer. Root apices were dissected on a glass slide and digested in a drop of enzyme mix (1% (*w*/*v*) cellulase, 0.5% (*w*/*v*) cytohelicase, 1% (*w*/*v*) pectolyase in PME) for 1 h at 37 °C. After three washes with PME 1X root apices were squashed gently between the slide and a coverslip, and frozen in liquid nitrogen. After removal of the coverslip and drying of the slides overnight, EdU revelation and Hoechst counterstaining were performed following the kit instructions (Thermo Fisher Scientific, Waltham, MA, USA). The percentage of EdU positive nuclei was plotted as a function of time. The percentage of EdU positive nuclei increases linearly with time and follows an equation that can be written as y = at + b where y is the percentage of EdU positive nuclei and t is time. Total cell cycle length is estimated as 100/a, and S phase length is b/a.

### 4.6. Modelling

YASARA was used to visualize the structural model of human polymerase α, which was downloaded from the Protein Data Bank (PDB) with ID number “5IUD”.

## Figures and Tables

**Figure 1 ijms-22-09409-f001:**
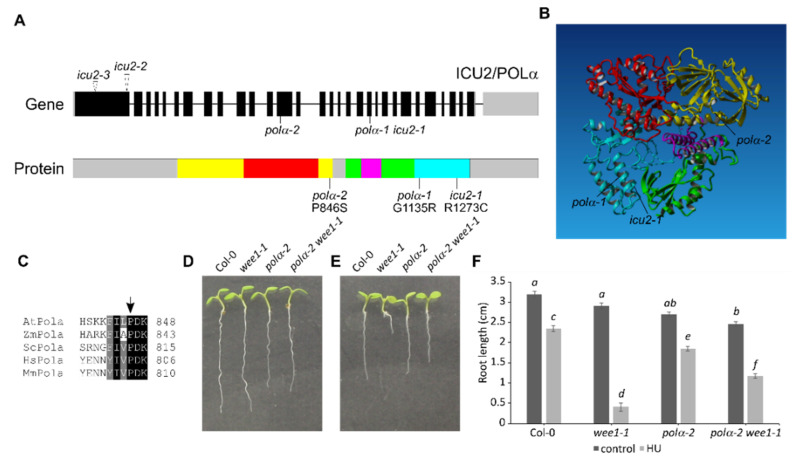
Characterization of the *polα-2* mutation. (**A**) Top, representation of the *Arabidopsis ICU2/Polα* genomic locus and positions of all known mutant alleles. Bottom, representation of the protein domains and location of the known mutations. Yellow, N-terminal domain; red, exonuclease; green, palm; magenta, finger; cyan, thumb; grey, uncharacterized. (**B**) Model of the human polymerase α based on the structure of [24] with overlay of *Arabidopsis* mutated alleles. Colors are identical as in A. (**C**) Alignment of protein sequences of *Arabidopsis*, maize, yeast, human, and mouse polymerase α. Mutated position of *polα-2* is indicated by arrow. (**D**,**E**) Root growth of 7-day-old wild-type (Col-0), *wee1-1*, *polα-2*, and *polα-2 wee1-1* seedlings grown on control medium (**D**) or medium supplemented with 0.75 mM HU (**E**). (**F**) Quantification of root growth shown in (**D**,**E**). Data represent the mean ± SEM (*n* > 10 in three biological repeats). Significance was tested with mixed model analysis. The means with different letters are significantly different (*p* < 0.05).

**Figure 2 ijms-22-09409-f002:**
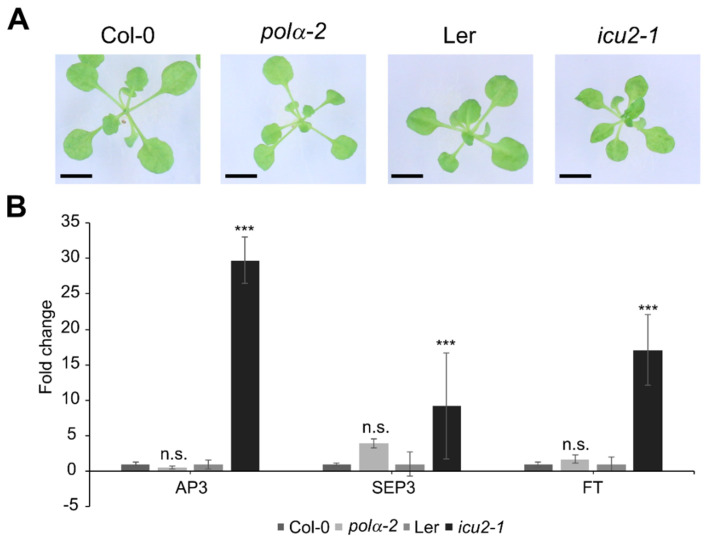
The *polα-2* mutation is phenotypically distinct from the described *icu2-1* mutation. (**A**) Rosette phenotype of 21−day−old wild−type (Col-0), *polα*−*2*, Ler, and *icu2*−*1* plants. Bars = 0.5 cm. (**B**) Gene expression levels of flowering genes in the first leaf pair of 21-day-old Col-0, *polα*−*2*, Ler, and *icu2*−*1*. Data represent the mean ± SEM (minimum 4 plants per repeat in 3 biological replicates), expression levels were normalized to the respective WT ecotype. Significance was tested with mixed model analysis. n.s., not significant; *** *p* < 0.001.

**Figure 3 ijms-22-09409-f003:**
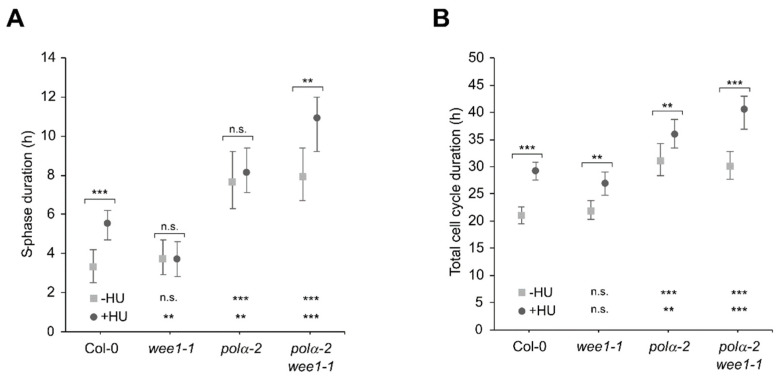
Cell cycle parameters of Col-0, *wee1-1*, *polα-2*, and *polα-2 wee1-1* root tip cells under control conditions and treated with HU. (**A**,**B**) S-phase (**A**) and total cell cycle (**B**) duration were measured using a time course of EdU staining according to the protocol of Hayashi et al. [25]. Data represent the mean ± 95% confidence intervals (n.s., not significant; ** *p* < 0.01; *** *p* < 0.001). For S-phase duration, comparisons were tested using ANOVA with Tukey correction. Total cell cycle duration was tested using ANOVA with F-tests to statistically test the equality of the means (*n* > 5). As inset, significance is depicted for each treatment of the comparison with Col-0.

**Figure 4 ijms-22-09409-f004:**
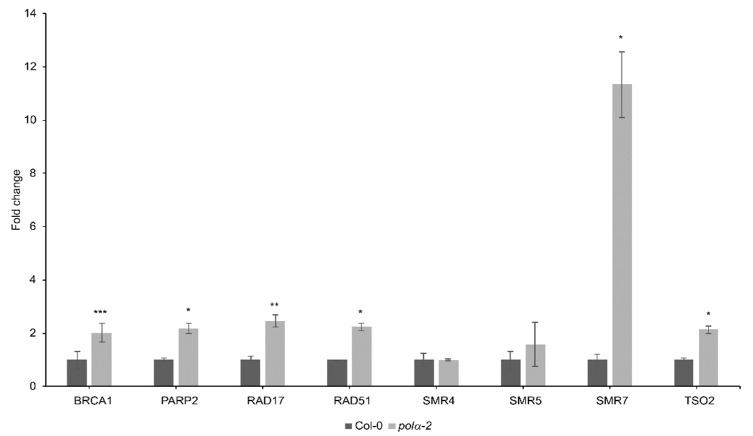
Analysis of DNA damage response genes in the *polα-2* mutant. Transcript levels of several DNA damage-related genes were measured in root tips of 7-d-old seedlings by qRT-PCR. Data represent the mean ± SEM (*n* = 3, minimum 100 root tips per repeat). Asterisks indicate statistical significance compared to Col-0 based on Student’s t-test; * *p* < 0.05; ** *p* < 0.01; *** *p* < 0.001.

**Figure 5 ijms-22-09409-f005:**
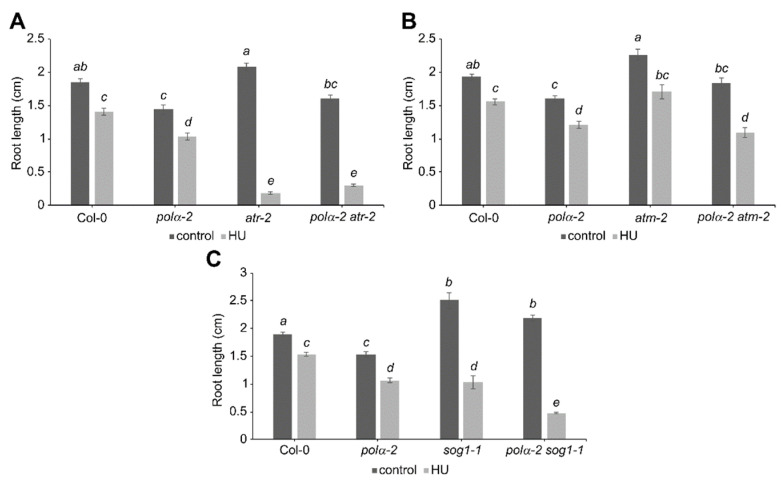
The *polα-2* mutation does not confer tolerance to replication stress in other DDR regulator mutant lines. Quantification of root growth of 7-day-old (**A**) wild-type (Col-0), *polα-2*, *atr-2*, and *polα-2 atr-2* seedlings, (**B**) wild-type (Col-0), *polα-2*, *atm-2*, and *polα-2 atm-2* seedlings, or (**C**) wild-type (Col-0), *polα-2*, *sog1-1*, and *polα-2 sog1-1* seedlings grown on control medium or medium supplemented with 0.75 mM HU. Data represent the mean ± SEM (*n* > 8). Significance was tested with mixed model analysis. The means with different letters are significantly different (*p* < 0.01).

**Figure 6 ijms-22-09409-f006:**
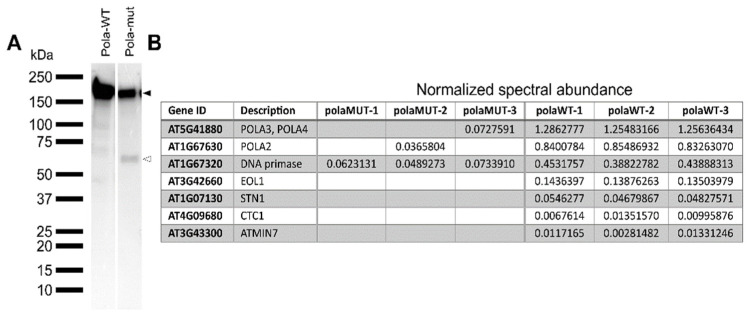
Identification of interactors of the wild-type and mutated form of DNA polymerase α. (**A**) Western blot of total protein extract using a peroxidase anti-peroxidase soluble complex antibody to detect the TAP fusion protein. Total protein was extracted from *Arabidopsis* PSB-D cell cultures transformed with N-terminally GSRhino-tagged WT (Polα-WT) and mutated (Polα-mut) form of DNA polymerase α. Extracts were run on the same gel in non-adjacent lanes. The black and white arrowheads indicate the full length and truncated protein form, respectively. (**B**) Proteins identified through tandem affinity purification of the constructs mentioned in (**A**). Values indicated are normalized spectral abundance factors, which are then normalized against the values of the bait protein, that was present in all samples. The three columns of each construct indicate three independent pulldowns of the protein. POLA3, POLA4: polymerase α primase subunit; POLA2: polymerase α regulatory subunit; EOL1: ENHANCER OF LHP1; STN1: SUPPRESSOR OF CDC13 HOMOLOG; CTC1: CST TELOMERE REPLICATION COMPLEX COMPONENT 1; ATMIN7: HOPM INTERACTOR 7.

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
