# Peer review of "A Mutation in DNA Polymerase α Rescues WEE1KO Sensitivity to HU"

_ijms, 2021, doi:10.3390/ijms22179409_

Round 1
Reviewer 1 Report
Eekhout et al describe the phenotype of a previously identified missense mutation in polymerase alpha (pol2) and its effect in combination with another mutation in wee1, a gene necessary to survive replication stress in Arabidopsis thaliana. Authors show that in the double mutant (pol2 and wee1) growth length observed in wee1 is partially recovered, and that rosette phenotype I pol2 is similar to the wildtype. They also reveal some differential expression of flowering genes comparing wee1 and the double wee1-pol2, and that the S-phase is prolonged for several hours even in the absence of the drug HU. Expression of genes involved in DNA repair seems to be affected two-fold and remarkably one of the tested gene, SMR7, involved in DNA damage response was upregulated for more than an order of magnitude, when compare wt with pol2 mutant. Finally, the authors identified differences in the proteins associated to the wt and to the pol2 mutant.
General comments:
The manuscript reads well and is logically presented, in general. Eukaryotic DNA replication is a fundamental topic in biology so the manuscript could attract many readers. Authors highlight in the introduction an interesting point by bringing the connection between pol alpha and epigenetic inheritance in plants What is the status in other organisms like mammals? Or yeast? Unfortunately, the topic is not brought back in the discussion or connected somehow to their own results. Are they planning to look into this? do they have suggestions for addressing the topic experimentally? Is this mutation related somehow to interactions with some chromatin remodelers? Please elaborate.
If space permitted, the abstract would benefit with a brief explanation (short sentence) explaining the connection between wee1 and CDK. The claim of the stability of the protein needs to be reexamined in light of the experimental results, see below.
As authors describe the phenotype as different and distinct from the pol1 and icu2-1 mutations, some of the results would be more informative by including a side-by-side comparison with pol2.
Figure legends and titles could be more descriptive an in several occasions, panels are referred in text not in the expected order. For example, Figure1C is mentioned before Figure1 B see below
Specific comments:
Page3 Line 100: when referring to a conserved domain authors show in Figure1c a limited homology by comparing 10 amino acids. Is the domain the N-terminal? if so, maybe better to show the full alignment and highlight the mutation site in the ‘I-PDK’ sequence
Page3 Figure 1 Panel A describe grey color in the legend. Panel B would be clearer with a zoom in the mutation region. Panel F ‘n>10’ does it mean more than 10 samples per condition? Clarify
Page3 line 120 CDS spell out
Page4 Figure 2 Panel A as a comparison would be informative to see side by side this phenotype for icu1-2 and /or pol1 as the authors want to make the point of being different. Is this phenotype analyzed in the presence or in the absence of HU? A brief indication in the text, and not only in the x axis of panels B and C, of the proteins AP1, AP3, FT, SEP3 and WUS would inform better the reader.
Page 4 line 147. Could the authors clarify the sentence: ‘ in the first leaf of 21 DAS wee1-1’
Page4 line 149. If significance was tested what was the result? Is there any significant difference between the gene expression?
Page 4 line 157: why the extended S-phase alone cannot explain the 10-hour increase? Please clarify
Page 5 line165 is the impairment of pola by mutation in pol1 and icu2 also leading to wee1 independent S-arrest or, is it exclusive of pol2?
Page5 line182 Upregulation of SMR7 seems remarkably high. However, there is little mention to this result. Is it something expected or normal in this type of mutants? Has been observed something similar in the other polalpha mutants pol1 and icu2? Is it known the relation between pol alpha and SMR7 , what is SM7 specifically doing?
Page6 line 185’ Figure legend title is not clear
Page6 Figure5: As Col-0 and pola2 data seems to be same, authors may consider simplifying the figure to one single panel including atr2, atm2 and sog1. Why the root length of the control and pol2 is shorter than in the experiment shown in figure 1 panel F?
Page7Figue6. Panel A. it is not specified what type of gel is shown (dPAGE %? Coomassie staining? Western blot? If so, what antibody was used?) or what samples where loaded (total cell extract, output from the pull down). In panel B same figure how many significant figures are relevant here? Is 0.0546277 (polWT1) any different form 0.04679867 (polWT2) in the interaction with STN1?
Author Response
See attached report

Reviewer 2 Report
In “A mutation in DNA polymerase α rescues WEE1KO sensitivity to HU” Thomas Eekhout and colleagues show how wee1 mutant is hypersensitive to replication stress and how polα-2 mutant, mutated in the catalytic subunit of DNA polymerase α, acts as a suppressor mutant of wee1, while no rescue of root growth under replication stress can be seen in polα-2 mutant lines together with atm-2, atr-2 or sog1-1.
The data in the paper are generally inconclusive or presented without a proper explanation of the phenomena observed. The statistical analysis is quite difficult to grasp at first glance. Most of the conclusions are based of root growth experiments, without any additional data obtained from different types of analysis. The authors are willing to explore the effect of DDR proteins combined with Pola-2 mutant, but fail in answering the questions in a cell molecular way.
Introduction:
- the authors focus their attention on mammalian cell proteins, while their paper is about plant cells. Although a few points could be useful for the general audience to understand, they should focus more on their field of research.
- The epigenetic modifications described are a little vague, few more details would be appreciated. Are there any studies regarding the effect of mono- or di-methylation in histones?
Results:
- Polα alone in control medium affects the root growth more than wee1-1 alone. A sentence to explain this will be useful. Also, wee1-1 root in Figure 1D looks significantly longer than control, but this does not reflect the data in Figure 1F.
- According to the authors “flowering genes are not significantly upregulated in its leaves (Figure 2B) or full rosette (Figure 2C)”, but the data regarding the polα-2 mutant are absent. Needs clarification.
- Still in Figure 2B, FT and WUS levels are significantly lower, although possibly not statistically significant, contradicting once again the statement in line 141-142.
- The authors states that “The extended S-phase alone cannot explain the 10 hour increase in total cell cycle duration, indicating the activation of a second cell cycle arrest, possibly in the G2/M-phase (Figure 3B)”. Although this is a reasonable explanation, no data are presented to support this statement, only a reference.
- The control in Figure 5 are quite variable among treatment. Assuming the control root length as 100%, it seems the pola-2 treatment has no difference compared to the control, but the double mutant might rescue the root length (when compare to its relative control) if compared to the atr-2 pair.
- Protein expression in Figure 6A indicates a much higher protein level in the mutant compared to the WT. Also, a WT band is clearly detectable in the mutant sample. Was this observed in any sample analysed? What is the ratio WT:MUT protein in the PolA-mut samples? Is the WT still catalytically active?
Discussion:
- In line 261, the authors state epigenetic modifications as general events without going into more needed details.
- In general, the discussion is poorly referenced.
Materials and methods:
- Adequate
Author Response
See attached document

Round 2
Reviewer 2 Report
NA
Round 3
Reviewer 2 Report
Due to the lack of new data presented regarding a deeper molecular biology investigation, as requested, this reviewer does not accept this manuscript for publication